# MaNtLE: A Model-agnostic Natural Language Explainer

**Rakesh R Menon**    **Kerem Zaman**    **Shashank Srivastava**

UNC Chapel Hill

{rrmenon, kzaman, ssrivastava}@cs.unc.edu

## Abstract

Understanding the internal reasoning behind the predictions of machine learning systems is increasingly vital, given their rising adoption and acceptance. While previous approaches, such as LIME generate algorithmic explanations by attributing importance to input features for individual examples, recent research indicates that practitioners prefer examining *language explanations that explain sub-groups of examples* (Lakkaraju et al., 2022). In this paper, we introduce **MaNtLE**, a model-agnostic natural language explainer that analyzes a set of classifier predictions and generates *faithful natural language explanations* of classifier rationale for structured classification tasks. **MaNtLE** uses multi-task training on thousands of synthetic classification tasks to generate faithful explanations. Our experiments indicate that, on average, **MaNtLE**-generated explanations are at least 11% more faithful compared to LIME and Anchors explanations across three tasks. Human evaluations demonstrate that users predict model behavior better using explanations from **MaNtLE** compared to other techniques.[1]

## 1 Introduction

The increasing adoption of black-box machine learning models across various applications (Shi et al., 2022; Dada et al., 2019) has led to a pressing need for human-understandable explanations of their decision-making process. While such models may yield high predictive accuracies, their underlying reasoning often remains opaque to users. This lack of transparency is a critical barrier to their adoption in critical domains, such as healthcare, law, and medicine.

To interpret machine learning model decisions, previous research proposed techniques like feature importances (LIME, Ribeiro et al. (2016)), rule

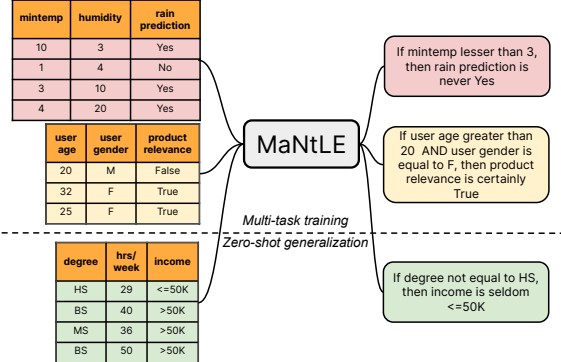

Figure 1: **MaNtLE** is a model-agnostic natural language explainer. Following massive multi-task learning, **MaNtLE** can generate explanations of decision-making rationale for new classifiers and datasets.

lists (Anchors, Ribeiro et al. (2018)), and model-generated explanations (Rajani et al., 2019; Narang et al., 2020). However, these explanations focus on model behavior at the level of individual examples rather than subgroups, which makes it hard for designers to identify systemic problems and refine models. Recent studies indicate that this limitation contributes to practitioners' reluctance to use machine learning systems for critical applications (Lakkaraju et al., 2019, 2022).

In this paper, we introduce **MaNtLE**, a model-agnostic natural language explainer that analyzes multiple classifier predictions and generates *faithful natural language explanations* of the classifier's reasoning for structured classification tasks, as depicted in Figure 1. The goal of **MaNtLE** is to explain the rationale of classifiers on real-world tasks. To develop **MaNtLE**, we fine-tune a `T5-Large` model on thousands of synthetic classification tasks, each paired with natural language explanations, in a multi-task learning setup following recent research (Wei et al., 2022; Sanh et al., 2022; Mishra et al., 2022; Menon et al., 2022). In §3.5, we discuss inference procedures to improve explanation quality and adapt the model trained on

---

[1] Code and pre-trained models are available at: https://github.com/rrmenon/MaNtLE

synthetic data for real-world tasks.

We test **MaNtLE** explanations on real-world tasks by assessing their ability to aid explanation-guided zero-shot classifiers in the CLUES-Real benchmark (Menon et al., 2022). Our results show that **MaNtLE** explanations are as helpful as human-written explanations in guiding classifiers (§5.2). We also compare the faithfulness (Jacovi and Goldberg, 2020) and simulatability (Hase and Bansal, 2020) of explanations generated by **MaNtLE**, LIME, and Anchors for four classification techniques on three real-world tasks (§5.3). When the number of available examples is comparable, **MaNtLE** explanations are, on average, **37% more faithful** than LIME and **11% more faithful** than Anchors.

In user studies (§5.4), we observe that users without machine learning expertise **prefer explanations from MaNtLE** over attribution-score-based explanations by LIME (overall preference of 3.44 vs 2.16 on a five-point Likert scale; $p < 0.001$ t-test). Further, users can predict model behavior better using **MaNtLE** explanations in **at least** 25% **more instances** than LIME and Anchors for the adult dataset (Dua and Graff, 2017). Our results corroborate the findings of Lakkaraju et al. (2022) on the need for providing natural language explanations that explain subgroups of examples to help practitioners interpret ML models. We find that increasing the number of examples accessible by **MaNtLE** can enhance explanation quality, highlighting possibilities for using the model in resource-rich settings. Finally, we analyze the contributions of the number of multitask-training tasks used and model size on the quality of **MaNtLE**'s generated explanations.

In summary, our contributions are:

- We develop **MaNtLE**, a model-agnostic natural language explainer that generates faithful language explanations of classifier rationale.

- We demonstrate the efficacy of **MaNtLE** by (1) comparing the classification-utility of explanations with human-written explanations on the CLUES-Real benchmark, and (2) evaluating the faithfulness and simulatability of explanations compared to popular approaches.

- We show that human users predict model behavior better with **MaNtLE** explanations compared to LIME and Anchors. Human users also rate **MaNtLE** explanations as better in understandability, informativeness, and overall preference.

- We analyze factors contributing to **MaNtLE**'s performance and outline opportunities for improving explanations.

## 2   Related Work

**Explainability methods.** Previous research has proposed methods for understanding model rationale, which can be broadly categorized into feature attribution and language explanation methods.

Feature attribution techniques (Kim et al., 2018; Lundberg and Lee, 2017; Sundararajan et al., 2017) provide information about how models use certain features to derive classification decisions, utilizing methods such as sparse linear regression models (Ribeiro et al., 2016, LIME) and rule lists (Ribeiro et al., 2018, Anchors). These methods have two shortcomings: (a) they explain model behavior in a local region around an example, and (b) they generate explanations by solving a search problem over a set of instances from the original training data or by generating perturbations around an example (typically, $\sim 1000$ examples). However, gaining access to model predictions on thousands of examples can be resource-intensive or financially unviable for large models, such as GPT-3 (Brown et al., 2020).

Recent works like CAGE (Rajani et al., 2019) and WT5 (Narang et al., 2020) explore natural language explanations for language understanding tasks. However, these explanations are specific to individual examples and focus on improving classification performance rather than interpreting model behavior. Moreover, training these models often demands a significant number of human-written explanations, which can be impractical. Our work diverges from this line of research as we seek to explain model behavior rather than improve classification models, using only a few examples.

**Multi-task Training of Language Models.** Large language models (LLMs) pretrained on large text corpora have shown impressive performances on various tasks (Brown et al., 2020; Scao et al., 2022). Some recent works (Wei et al., 2022; Sanh et al., 2022) have explored multitask training of language models on labeled datasets paired with natural language instructions to enhance zero-shot generalization, a procedure called instruction fine-tuning. In contrast, **MaNtLE** generates explanations when prompted with feature-prediction pairs.

## 3 MaNtLE

### 3.1 Problem Setup

MaNtLE takes as input a set of input-prediction pairs, $\{(X_i, y_{\theta_t, i})_{i:1 \rightarrow N}\}$, from a black-box classifier, $\theta_t$, trained for a structured classification task $t$. The output from MaNtLE is an explanation, $e \in \mathcal{E}$, where $\mathcal{E}$ is the set of all possible natural language explanations. An explanation is a natural language statement that describes the behavior of the classifier in predicting labels $y_{\theta_t, 1:M}$ for the inputs $X_{1:M}$. Figure 1 shows illustrative examples of explanations generated for three classifiers and tasks.

### 3.2 Model

We frame the task of explanation generation from input-prediction pairs as a sequence-to-sequence problem. Framing the task as a sequence-to-sequence problem enables fine-tuning of pre-trained language models to generate language explanations. We initialize MaNtLE using T5-Large (Raffel et al., 2020) [2] language model. We linearize input-prediction pairs as text and predict text explanations using our model. Our input-linearization strategy converts a set of $k$ examples into text with an additional prompt, explanation: <extra_id_0>. Figure 2 illustrates this linearization and encoding process with two examples. On the decoder side, we begin prediction from the <extra_id_0> sentinel token, matching the span-prediction task that T5 is optimized for.

| a | b | lbl |
|---|---|---|
| yes | 2 | True |
| no | 2 | False |

*Per-example Text Encoding:*
a equal to yes. b equal to 2. lbl equal to True.
a equal to no. b equal to 2. lbl equal to False.

*MaNtLE Input:*

**example 1**: a equal to yes. b equal to 2. lbl equal to True.
**example 2**: a equal to no. b equal to 2. lbl equal to False.
**explanation**: <extra_id_0>

Figure 2: MaNtLE's strategy for linearizing structured data to text.

### 3.3 Multi-task Training

To train MaNtLE, we use massive multi-task training following recent advancements in this area (Wei et al., 2022; Sanh et al., 2022; Mishra et al., 2022). This approach equips MaNtLE with the ability to generate explanations for any classifier without the need for additional fine-tuning. The degree of generalization achieved by the model depends on the

number and diversity of tasks used during multi-task training. However, obtaining language explanations for thousands of classifiers for training purposes is infeasible. To address this, we adopt a strategy proposed by Menon et al. (2022) and generate synthetic classification tasks programmatically, where the tasks have known natural language explanations and the examples are used to predict these explanations. To diversify the set of tasks, we generate tasks with explanations that include a variety of logical operations in addition to explanations conditioned on a single feature, e.g., *'If fruit color is red, then apple'*. We mirror Menon et al. (2022) in varying synthetic tasks and explanations in terms of the logical operations present, including *conjunctions*, *disjunctions*, *negations*, and *quantifiers*. We refer the reader to §A.3 and Table 4 for information on the task variations/complexities. These variations are based on prior research, which explores the linguistic elements used by humans to communicate task intents through language (Chopra et al., 2019).

Additionally, we assume that users are interested in understanding the rationales for a specific label of a classifier at a time. Thus, we re-frame all examples as binary classification examples for MaNtLE, where the label of interest is maintained, as {label} say, and the remaining labels are relabeled as "not {label}".

### 3.4 Training Details

We perform multi-task training for MaNtLE on $\sim$ 200K synthetic datasets that have five features in all tasks covering a wide range of explanation types (§3.2). We cap the maximum number of tokens to 1024 and 64 tokens, respectively, for the input and the output. This corresponds to packing between 10-12 examples in the input when generating explanations for classifier rationale. We optimize the model to maximize the likelihood of ground-truth explanations given the set of examples. Additionally, since MaNtLE derives explanations by extracting patterns in examples from both classes ({label} and not {label}), we ensure that at least 10% of the input examples belong to each of the two classes. The model is trained for 2 million steps with a batch size of 4 using AdamW (Loshchilov and Hutter, 2019), a learning rate of 1e-5, and a linear decay schedule with a warmup of 50K steps.

After training, we select the best model check-

---

[2] Due to computational constraints, we do not experiment with larger models (T5-3B or T5-11B).

point using the highest score (sum of all metrics in §3.6) on the validation sets of 20 randomly selected synthetic tasks from those used during training.

## 3.5 Inference Techniques

For inference, we experiment with three decoding approaches. The first is greedy decoding, where we generate the most likely explanation from **MaNtLE** conditioned on the input. The second approach aims to improve explanations by sampling multiple candidates and returning the explanation that most faithfully explains model behavior on the inputs (see §3.6 for the definition of faithfulness). For this, we use beam-search, with beam size 20, and generate 20 explanations from **MaNtLE**, following which we return the most faithful explanation. Assessing beam-search generations, we found that sequences often repeated the same feature. To diversify explanations, we develop PerFeat decoding, our third decoding approach. Here, we prompt the **MaNtLE** decoder with 'If {feature_name}' to generate sequences for each feature and return the most faithful explanation.

## 3.6 Evaluation Metrics

To evaluate the generated language explanations, we use BERTScore (Zhang* et al., 2020), *ROUGE* (Lin, 2004), and *BLEU* (Papineni et al., 2002). However, these metrics capture surface-level similarities between the generated explanation and a reference. In our scenario, we want to ensure that the explanations are faithful to the input and that users can simulate model behavior on new examples based on the explanation. We measure *faithfulness* by using the explanations to make predictions on the inputs, $X_{1:N}$, and evaluating how often the labels suggested by the explanations match with the original predictions from the classifier in question, $y_{1:N}$.[3] Similarly, to measure *simulatability*, we use the explanations to make predictions on unseen examples from the task, $X'_{1:M}$, and assess their alignment with the ground-truth labels, $y'_{1:M}$. We use a semantic parser to parse explanations generated for unseen synthetic and real-world tasks.

## 4 Explaining Datasets with **MaNtLE**

### 4.1 Synthetic Datasets

Following training, we evaluate **MaNtLE** by generating explanations for 20 synthetic datasets from

the different task complexities described in §A.3 (presence/absence of *conjunctions*, *negations*, and *quantifiers*; Table 4). These datasets were not used during training and are therefore considered *unseen* tasks. We use greedy decoding to generate the explanations from **MaNtLE**.

Example generations in Table 1 indicate that while we train **MaNtLE** equally across all complexities, the generations are biased towards using quantifiers in the explanations. Consequently, surface-level metrics, such as BLEU, are highest for the generated explanations in the quantifier tasks category (see Figure 3a). The generated explanations seldom contain conjunctions leading to lower faithfulness and simulatability measures on datasets that contain conjunctive explanations in Figure 3. Further, generated explanations never have negations in the label, i.e., no 'not {label}' explanations. Hence, the faithfulness and simulatability are lower than the no-negation datasets (Figure 3; second and fourth bars in each of the four sets).

### 4.2 Real-world Datasets: CLUES-Real

We investigate the efficacy of **MaNtLE** explanations compared to human-written explanations for aiding in classifying examples from datasets in the CLUES-Real benchmark (Menon et al., 2022). Specifically, we evaluate the accuracy of LaSQuE (Ghosh et al., 2022), a recent explanation-guided classifier for the benchmark,[4] using both explanations across three binary classification tasks from the benchmark's zero-shot evaluation set. Explanations justifying the labeling rationale for datasets in CLUES-Real were developed by individuals that were shown a few examples. Hence, this benchmark provides an ideal setting for evaluating **MaNtLE** explanations, which are also generated based on patterns observed in a few examples.[5]

Results in Table 2 show the the performance of LaSQuE with the best **MaNtLE** explanations is able to get within 7% of the performance with crowd-sourced explanations across tasks. Additionally, when optimizing for faithfulness, diversifying the pool of candidate explanations is helpful as indicated by the consistent performance improvement of PerFeat decoding over beam search.

---

[3] This approach is also referred to as *fidelity* in the Explainable AI literature (Jacovi and Goldberg, 2020).

[4] FLAN-T5-XL (Chung et al., 2022), an effective model at learning from instructions, underperformed LaSQuE here.

[5] In the implementation, we test the performance of the top-10 generated **MaNtLE** explanations (seeing 10 input examples) with 10 crowd-sourced explanations for each task from CLUES-Real.

| Task Complexity | Ground-truth Explanations | MaNtLE Explanations |
|---|---|---|
| simple | If pdsu lesser than or equal to 1014, then no | If pdsu not greater than 1020, then it is certainly no |
| | If vpgu equal to antartica, then blicket | If vpgu equal to antartica, then it is definitely blicket |
| quantifier | If twqk equal to no, then it is seldom fem | If twqk equal to no, then it is seldom fem |
| | If bgbs not equal to 4, then it is certainly 2 | If bgbs equal to 4, then it is seldom 2 |
| conjunction | If aehw equal to no AND hxva equal to africas, then tupa. | If hxva equal to africas, then it is definitely tupa |
| | If kjwx greater than or equal to 18 OR bzjf greater than 1601, then it is definitely 1. | If kjwx not lesser than 19, then it is likely 1 |

Table 1: Explanations generated by MaNtLE for unseen synthetic tasks for different task complexities.

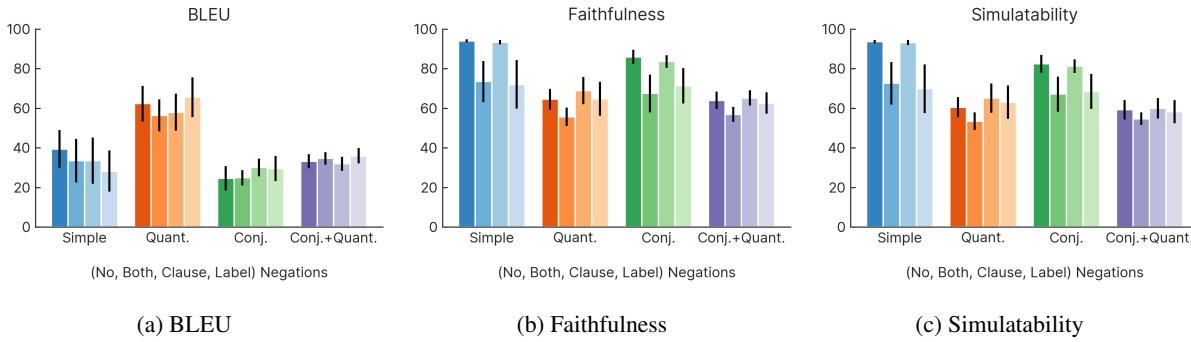

(a) BLEU

(b) Faithfulness

(c) Simulatability

Figure 3: Results on unseen tasks of different complexities (negations, quantifiers, conjunctions). These are numbers averaged over 20 datasets per task category. Error bars indicate standard deviation.

| Task | MaNtLE Decoding Strategy | Accuracy | Human Explanation Accuracy |
|---|---|---|---|
| banknote authentication | Greedy | 50.6 | |
| | Beam | 52.0 | 56.4 |
| | PerFeat | **52.4** | |
| indian liver patient | Greedy | 44.6 | |
| | Beam | 40.5 | 48.6 |
| | PerFeat | **51.4** | |
| tic-tac-toe endgame | Greedy | **63.0** | |
| | Beam | 56.8 | 66.1 |
| | PerFeat | 57.8 | |

Table 2: Simulatability of MaNtLE explanations as measured by LaSQuE (Ghosh et al., 2022). Accuracy measures LaSQuE's ability to predict ground-truth labels.

We provide qualitative examples of generated explanations for different tasks in Table 5. While MaNtLE can recover some crowd-sourced explanations, we also observed hallucinations and a lack of numerical understanding leading to errors in explanations. We leave these errors for future studies to investigate and address.

## 5 Interpreting Classifiers with MaNtLE

In this section, we compare the quality of MaNtLE explanations to two popular explanation methods in LIME and Anchors using simulated (§5.3) and human user studies (§5.4). Before the experiments,

we briefly describe the baselines (§5.1) and datasets (§5.2) that we use in our experimentation.

### 5.1 Baseline Explanation Methods

**LIME.** Ribeiro et al. (2016) approximates model behavior locally by fitting a linear model to the outputs of the black box model on perturbations sampled from a distribution around an input. The linear model output is the probability of an example belonging to the same class as the input used for constructing the explanation. The quality of LIME explanations depends on the number of perturbations sampled around an input. Typically, LIME uses 5000 perturbations to fit a good linear model and generate explanations for tabular datasets.

**Anchors.** Ribeiro et al. (2018) proposed a technique for building rule lists that predict model behavior with high precision, i.e., the label mentioned in the explanation matches the model prediction with a high likelihood when the rules apply. To achieve high precision, Anchors often sacrifices explanation coverage, i.e., the number of instances where the rules apply may be very limited. Anchors employs a PAC (probably approximately correct) learning approach to generate a rule list based on samples from a distribution around an input. Unlike LIME which uses input perturbations, Anchors uses training set samples of the black-box model

that is close to the input for its construction.

**On the omission of WT5.** To evaluate the efficacy of natural language explanation generation methods, it is common to compare with established works such as WT5 (Narang et al., 2020) and CAGE (Rajani et al., 2019). Nonetheless, we have chosen to abstain from comparing our approach with these works in this section for the following specific reasons:

- Natural Language Explanations (NLE) baselines, such as CAGE or WT5, jointly predict labels and supporting explanations to enhance classification through knowledge induction.

  In contrast, `MaNtLE` and methods like LIME or Anchors operate independently from label-predicting classifiers, providing post-hoc explanations. `MaNtLE` can be applied in a model-agnostic manner for any classifier, focusing on explaining classifiers rather than enhancing their performance. This distinction explains the absence of LIME or Anchors as baselines in prior NLE works (Rajani et al., 2019; Narang et al., 2020).

- CAGE and WT5 explain single-example predictions, while `MaNtLE` elucidates patterns within sets of examples. A fair comparison with `MaNtLE` necessitates a method capable of aggregating individual predictions.

## 5.2 Datasets and Models

Following previous work (Ribeiro et al., 2018), we perform experiments for three structured classification tasks. The first is the `adult` dataset from the UCI-ML repository (Dua and Graff, 2017), where the objective is to predict the income level of a person (more or less than $50,000$). The second dataset used is the `recidivism` dataset (Schmidt and Witte, 1988), where the task is to predict if a convict is likely to re-commit crimes. The third dataset is the `travel-insurance` dataset (Tejashvi, 2019) obtained from Kaggle, where the task is to predict if a traveler will purchase travel insurance.

In all experiments, we use five features from the available set of features for each dataset.[6] To ensure consistency, we follow the data-processing scheme and model architectures used in Ribeiro et al. (2018). For each dataset, we train and explain

---

[6] For each dataset, we use the top-5 features that maximize information between labels and examples in the training set.

four classifiers: logistic regression, decision trees, neural networks, and gradient-boosted trees.

We report the faithfulness and simulatability of explanations generated by different methods. Here, simulatability is measured as the proportion of test set examples for which the model prediction matches the label in the explanation.

## 5.3 Automated Evaluation

In this section, we conduct simulated user studies to compare the effectiveness of LIME, Anchors, and `MaNtLE` in generating explanations for classifiers.

**Setup.** For each classifier-dataset pair, we sub-sample 100 random subsets from the validation set, each with 10 examples and the corresponding predictions from the classifier. Next, for each subset, we generate explanations using LIME, Anchors, and the different `MaNtLE` variants. For LIME and Anchors, we compute the submodular pick (SP) variant of the explanations (Ribeiro et al., 2016).

As mentioned in §5.1, LIME and Anchors need to sample examples or perturbations around the input to generate high-quality explanations. However, `MaNtLE` generates explanations without any additional information beyond the examples from the subset. To perform a fair evaluation, we report results for a budget-constrained setting, wherein methods can make a maximum of 15 classifier calls. This corresponds to performing 1 perturbation per example for LIME and using 5 training examples for Anchors. Budget-constrained scenarios form a realistic setting in the current landscape where black-box classifiers, like GPT-3, are expensive to query both monetarily and computationally.

**Results.** In Table 3, we see that LIME falls short of Anchors and `MaNtLE` variants on all classifier-dataset combinations, likely caused by the small number of perturbations available to LIME. Among different `MaNtLE` decoding strategies, the results suggest that faithfulness improves with better decoding strategies, with `MaNtLE`-PF having the best performance overall. Overall, we observe that across the three datasets, `MaNtLE`-PF is more faithful than LIME and Anchors by $37\%$ and $11\%$, respectively, highlighting the utility of our approach in this budget-constrained scenario.

To address scenarios where many examples are accessible cheaply, in §6.2, we explore ways to incorporate more examples to improve the quality of `MaNtLE` explanations.

| | explanation | lr faith | lr sim | dt faith | dt sim | nn faith | nn sim | xgb faith | xgb sim |
|---|---|---|---|---|---|---|---|---|---|
| adult | LIME | 51.3 | 50.2 | 53.4 | 49.8 | 53.9 | 50.8 | 52.8 | 50.0 |
| | Anchor | **80.2** | **70.9** | 57.8 | 52.9 | 57.3 | 50.7 | 55.3 | 52.1 |
| | MaNtLE | 56.3 | 49.2 | 55.8 | 49.2 | 56.2 | 49.6 | 57.1 | 49.3 |
| | MaNtLE-BS | 67.6 | 52.9 | 67.1 | 51.9 | 67.5 | 52.6 | 68.4 | 51.4 |
| | MaNtLE-PF | 74.3 | 57.4 | **74.4** | **54.6** | **75.1** | **56.5** | **75.0** | **55.9** |
| travel ins. | LIME | 53.0 | 50.6 | 54.9 | 54.0 | 52.6 | 51.3 | 53.0 | 50.4 |
| | Anchor | 60.6 | **69.6** | 73.5 | **71.7** | 61.3 | **69.2** | 45.5 | 51.1 |
| | MaNtLE | 58.7 | 61.1 | 56.4 | 63.0 | 57.5 | 60.0 | 55.1 | 52.8 |
| | MaNtLE-BS | **73.3** | 63.5 | 71.7 | 63.0 | **72.7** | 62.5 | 68.8 | **54.1** |
| | MaNtLE-PF | 72.6 | 60.2 | 71.7 | 60.5 | 72.1 | 60.3 | **71.8** | 53.4 |
| recidivism | LIME | 53.9 | 50.0 | 51.4 | 50.0 | 51.7 | 50.0 | 50.7 | 50.0 |
| | Anchor | 58.5 | **60.6** | **76.1** | **58.8** | **77.4** | **58.5** | **76.1** | **58.9** |
| | MaNtLE | 53.9 | 50.6 | 54.3 | 51.9 | 52.4 | 51.4 | 53.9 | 51.7 |
| | MaNtLE-BS | 69.7 | 51.3 | 69.1 | 52.8 | 68.8 | 52.7 | 69.1 | 52.5 |
| | MaNtLE-PF | **70.9** | 51.6 | 69.3 | 51.6 | 69.7 | 51.3 | 69.4 | 51.1 |

Table 3: Faithfulness and simulatability when executing different explanations for three datasets. Results are averaged over 100 runs (or subsets). **Bold** numbers indicate the best explanation for a particular classifier-metric pair. SP = submodular pick, BS = beam search, PF = PerFeat decoding.

## 5.4 Human Evaluation

In user studies, we measure the ability of users to interpret explanations and predict the behavior of models trained on the adult dataset. We use the full budget LIME and Anchors explanations and the **MaNtLE**-PerFeat explanations from the previous section. In a pilot study, we found workers had difficulty interpreting the meaning of different quantifiers. Hence, we post-process **MaNtLE** explanations by converting quantifiers to confidence values based on previous work (Menon et al., 2022).[7]

**Which explanations help users predict model behavior?** In Figure 4, we present the results of our study. We report the percentage of instances where the user predictions of model behavior improve, worsen, or remain the same on perturbed examples (perturbation) and test examples (simulation) after reviewing explanations, following the setup in Hase and Bansal (2020). 23 workers took part in this study conducted on Amazon Mechanical Turk and were compensated at $12/hr.

In our results (Figure 4), we found that users can predict model behavior more accurately in 46% of cases after reviewing **MaNtLE** explanations, compared to 15% for LIME and 19% for Anchors. Additionally, users were less likely to make more incorrect predictions of model behavior after viewing **MaNtLE** explanations, with only 19% of cases, compared to 38% for LIME and 31% for Anchors. Hence, our explanations are more reliable in helping users predict model behavior.

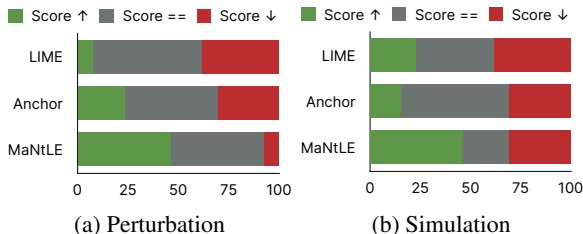

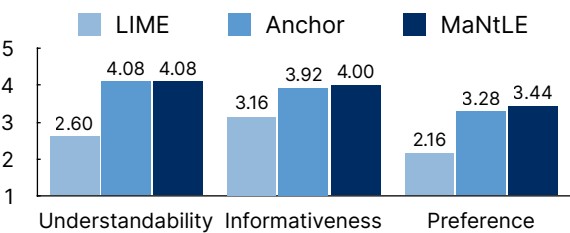

Figure 4: Percentage of instances where workers understanding of model behavior improved (green), declined (red), and did not change (gray) on reviewing different explanations for the adult dataset.

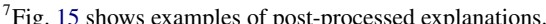

Figure 5: Preference for different explanation techniques by workers with at least undergraduate level education, as indicated by Likert ratings on a 1-5 scale.

**Which explanations would general practitioners prefer?** For this study, we recruited 25 participants on Prolific who are currently pursuing at least an undergraduate degree to rate explanations on a 1-5 Likert scale for understandability, informativeness, and overall preference.[8] We chose this demographic to reflect the expected diversity of industry experts' educational backgrounds likely to use explanations to better understand their systems.

Results in Fig. 5 indicate that workers struggled to comprehend attribution scores from LIME compared to **MaNtLE**. A paired-sample t-test for overall preference revealed significance (2.16 vs. 3.44; $p$-value $< 0.001$). In contrast, workers found Anchors informative, but their low coverage hindered their overall preference compared to **MaNtLE**.

Notably, workers prefer **MaNtLE** explanations for their clarity and simplicity, citing phrases such as *'clearly defines information'* and *'uses more layman terms'*. Nevertheless, some expressed concerns that the explanations were *'not descriptive enough'*, *'...semi-specific when compared to LIME'*. This suggests that explanations clarifying how each feature affects the classifier's decision-making process could improve user understanding and satisfaction.

---

[7]Fig. 15 shows examples of post-processed explanations.

[8]Fig. 14 shows how we define the scale for each property.

# 6 Analysis

In this section, we analyze **MaNtLE** based on two key aspects: (a) factors affecting the strong generalization of **MaNtLE**, and (b) stability of **MaNtLE** when the number of input examples is varied.

## 6.1 How does scale affect the performance of **MaNtLE**?

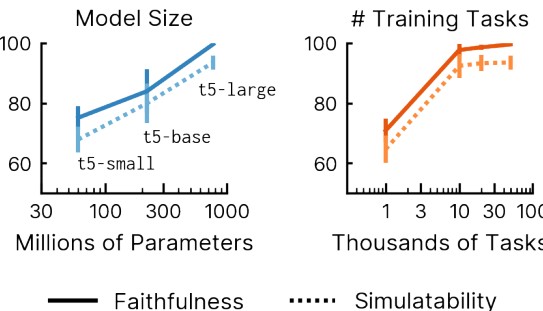

Figure 6: Faithfulness and simulatability performance of **MaNtLE** on held-out tasks with increase in the scale of model size (left) and dataset size (right). Error bars indicate standard deviation.

First, we evaluate the influence of model size and the number of training tasks on the quality of explanations generated by **MaNtLE**. For this, we create a synthetic benchmark consisting of 50K datasets, each with ground-truth explanations using conjunctions, which are challenging for **MaNtLE** to learn. For evaluation, 20 datasets from the benchmark are held-out, and we measure the faithfulness and simulatability of explanations generated by fine-tuned models on these datasets. We fine-tune T5 models using nearly all 50K datasets for model scale experiments, and vary tasks between 1K to 50K for task scale experiments (using a `T5-large` model here).

To study the effect of model scale, we fine-tune different variants of T5, ranging from `T5-Small` to `T5-Large`. As seen in Figure 6 (left), increasing the scale of models from `T5-Small` to `T5-Large` consistently improves both faithfulness and simulatability. Moreover, increasing the number of training tasks also enhances both metrics, as seen in Figure 6 (right). Notably, fine-tuning a `T5-Large` model on smaller number of tasks (1K) leads to poorer performance than a `T5-Small` model fine-tuned on larger number of tasks (50K), even when trained with the same hyperparameters for the same duration. Taken together, expanding multi-task training and increasing model sizes further could improve **MaNtLE**'s performance.

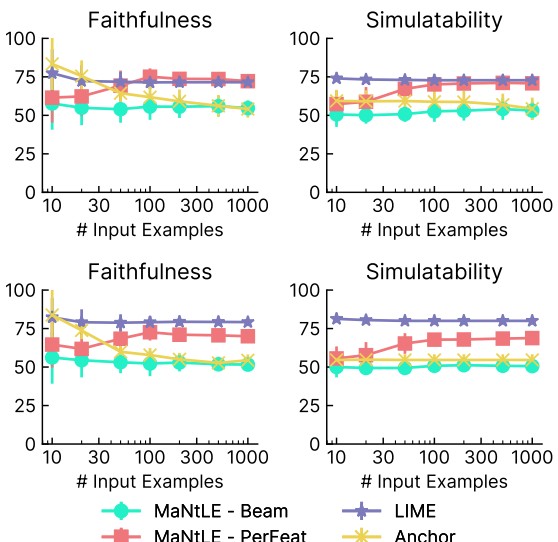

Figure 7: **Increasing number of input examples improves simulatability and faithfulness of `MaNtLE`-PF explanations.** Here, we compute metrics over 100 runs for two classifiers (*logistic regression – top row*, *decision tree – bottom row*) trained on the `adult` dataset. LIME and Anchors shown here are full-budget submodular pick variants. Error bars show standard deviation.

## 6.2 Can **MaNtLE** take advantage of more examples?

The maximum number of tokens allowed in the T5 encoder limits the input capacity of **MaNtLE**. This restricts **MaNtLE** when a large number of examples are available for the classifier being explained.

To overcome this challenge, we propose a technique that enables **MaNtLE**-Beam and **MaNtLE**-PF to handle a greater number of input examples. Our method involves dividing a set of $N$ available examples into eight subsets, each containing 10 examples. We utilize these subsets to generate explanations and select the explanation with the highest "simulatability" score among the $N$ examples as the best explanation. We evaluate the faithfulness and simulatability of our explanations using this approach to explain logistic regression and decision tree classifiers for the `adult` dataset.

In Figure 7, we see that our procedure enhances the faithfulness and simulatability of **MaNtLE**-PF. Additionally, the performance of **MaNtLE**-PF is comparable to the full-budget submodular pick variant of LIME, indicating that our approach enables **MaNtLE** to achieve comparable explanation quality when provided access to a large number of examples. However, the procedure does not yield improvements in the metrics for **MaNtLE**-Beam, implying that the diversity of generated explanations

is crucial. Expanding the number of subsets could further improve the performance of `MaNtLE`-PF. We leave this for future work to explore.

## 7 Conclusion

In this work, we introduce `MaNtLE`, a model-agnostic natural language explainer, that generates faithful explanations for structured classifiers. We use recent insights in massive multi-task training to train a model that can generate faithful classifier rationale. As `MaNtLE` can explain classifiers simply by inspecting predictions made by classifiers, it can be used in a model-agnostic fashion similar to popular explanation methods like LIME and Anchors. In simulation studies and human evaluations, we show that `MaNtLE` explanations are more faithful than LIME and comparable in faithfulness to Anchors on multiple datasets. Our work suggests the potential for natural language interfaces to enhance end-user trust and promote the safe usage of ML models in critical tasks by providing explanations of the classifier decision-making process. Future work can look to extend our work to develop "patches" (Murty et al., 2022) for improving classifiers, refining decoding techniques for more faithful and simulatable explanations, and integrating more complex reasoning patterns in generated explanations. Further, developing models that can generate language explanations for classifiers trained on other modalities, such as raw text and vision, is an interesting avenue for future work.

## Limitations

Our method is exclusively designed to explain classifiers operating on structured datasets. Utilizing `MaNtLE` for other input types, such as raw text and images, is out of the scope of this work.

Further, the number of examples that can be packed into the encoder of the `MaNtLE` is limited to 1024 tokens (limit of `T5` encoder). While we examine additional strategies to circumvent this issue, future work could look into additional methods for packing more examples into the input to improve explanation quality.

Additionally, the logic that can be represented by the outputs of `MaNtLE` is limited to the kind seen during training. Here, this implies that we may never observe explanations with nested conjunctions (three or more feature constraints combined). Future work can identify solutions to incorporate more complex reasoning in explanations. Integrat-

ing such reasoning without training `MaNtLE` from scratch is an intriguing avenue for future research.

## Ethics Statement

We perform experiments using a mixture of synthetically generated datasets as well as publicly available datasets from the UCI and Kaggle repositories. In human evaluations, workers were provided fair compensation over the federal minimum wage. Our work is a research prototype for an explanation method. If the inputs are aligned with the techniques mentioned in this work, we do not foresee risks for models (aside from hallucinations which have been discussed in the limitations section). However, `MaNtLE` is fine-tuned from the `T5` checkpoint, a pre-trained transformer model. Hence, it is possible that our model may exhibit unwanted biases that are inherent in pre-trained language models. This aspect is beyond the scope of our work.

## Acknowledgments

The authors would like to thank the anonymous reviewers for their suggestions and feedback on the work. This work was supported in part by NSF grant DRL2112635. The views contained in this article are those of the authors and not of the funding agency.

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

# A Experiment Details

Here we provide implementation details such as hyperparameters, hardware and software used for developing **MaNtLE** and running our experiments.

## A.1 Libraries

We use the HuggingFace library (Wolf et al., 2020) for all the transformer-based models. For T5 models, we experiment with t5-small, t5-base, and t5-large[9] across the main experiments and analyses. Our main experiment results are based on the t5-large model. Pre-trained checkpoints for these models are publicly available on the HuggingFace library. All models are coded in PyTorch 1.13.1 (Paszke et al., 2019).

## A.2 Pre-training Hyperparameters

We used the t5-large model and trained it using the AdamW (Loshchilov and Hutter, 2019) optimizer with a learning rate of $1e-5$ and weight decay of $1e-2$ for $2,000,000$ steps using the standard language modeling objective. Each gradient step is computed over a batch of 4 samples with no gradient accumulation steps. The maximum

---

[9]https://huggingface.co/t5-large

length of the input is clipped to 1024 tokens, which roughly corresponds to 10-12 input-prediction pairs being encoded in each sample, while we limit the decoder to generating 64 tokens since that was sufficient to generate the longest explanations from our training set. We chose the best checkpoint based on the performance over generation metrics as well as faithfulness and simulatability on 20 held-out validation tasks.

The model was fine-tuned using full precision on a single NVIDIA A100-PCIE-40GB GPU, 400GB RAM, and 40 CPU cores for $\sim 25$ days.

### A.3 Pre-training Datasets

Pre-training was performed using programmatically generated datasets whose explanations followed the if-then structure following Menon et al. (2022). We also utilize the different complexities described in this prior work, which enabled our model to perform diverse types of reasoning. Overall, there were 24 different complexities that varied by: (a) the presence of quantifiers in explanations, (b) the presence of conjunctions in explanations, and (c) the presence of negations in explanations. The quantifiers we adopt in this work, along with their values, follow from prior work in Srivastava et al. (2018). For conjunctions, we can have tasks with explanations that have nested conjunctions (AND-OR / OR-AND explanations) or simple conjunctions (AND / OR). For negations, there are more subdivisions based on the positioning of the negation. For example, if we have an explanation of the form, *'If a equal to 1 then yes'*, then a clause negation corresponds to an explanation of the form *'If a not equal to 1, then yes'*, and a label negation corresponds to *'If a equal to 1, then not yes'*. Hence, the presence of negations can vary by no negations, only clause negations, only label negations, and negations in clause+label. Overall, we have 2 (quantifier) $\times$ 3 (conjunctions) $\times$ 4 (negations) = 24 different complexities. We use $\sim 8,000$ tasks per complexity for training, which leads to the massive pre-training dataset of $\sim 200,000$ tasks. Table 4 concisely lists the template used for different task complexities.

Each synthetic task consists of five features and are sampled based on the synthetic templates in Menon et al. (2022). The range of values in our synthetic datasets thus follow the available range in CLUES. On the sampled datasets, we choose to convert feature names to random strings of four char-

| ztwh | hwyw | nwfh | zosp | ghcq | label |
|------|------|------|------|------|-------|
| no   | 7826 | 4    | 22   | 2    | **5** |
| yes  | 6668 | 5    | 51   | 5    | **5** |
| no   | 1201 | 3    | 32   | 1    | **Not 5** |

**Explanation:** If hwyw greater than 1528 , then 5

Figure 8: Example synthetic dataset used in our multi-task mixture to train `MaNtLE`.

acters. Early experiments indicated that reusing feature names across tasks with different classification logic can hurt model's ability to learn to generate explanations, hence prompting the random strings for feature names. We show an example of a sampled task for multi-task pre-training in Fig. 8.

### A.4 Real-world Task Explanation Generation

Owing to the pre-training procedure of `MaNtLE`, which involved feature names as single-word lowercase strings, it is essential that input to `MaNtLE` for real-world tasks is formatted in a similar fashion. Therefore, we transform all feature names from real-world tasks into a format that can be processed by `MaNtLE`, accomplished by eliminating spaces and converting all characters in the feature names to lowercase. To ensure that the generated explanations are understandable to humans, we perform a post-processing step to convert these lowercase feature names back into their original format. This process is crucial for achieving accurate explanations using `MaNtLE`.

### A.5 Model and Dataset Scale Analysis

For the model-scale analysis, we fine-tune from a `t5-large` checkpoint using the conjunction datasets. These models were trained using similar hyperparameters as that from pre-training. However, since the datasets only come from a single complexity, namely conjunctions, these models were trained for $200,000$ steps.

## B Extended Related Work

**Instruction Generation by Large Language Models.** Some recent works (Honovich et al., 2022; Singh et al., 2022) explore techniques to prompt LLMs to generate instructions based on a few examples from synthetic and real-world

| Task Complexity | Template | Example Explanations |
|---|---|---|
| simple | If {cond}, then {label} | If pdsu lesser than or equal to 1014, then no |
| conjunction | If {cond1} **AND/OR** {cond1}, then {label} | If aehw equal to no AND hxva equal to africas, then tupa |
| clause negation | If {feat_name} **not** equal to {feat_value}, then {label} | If bgbs not equal to 4, then 2 |
| label negation | If {cond(s)}, then **not** {label} | If aehw equal to yes, then not tupa. |
| clause+label negation | If {feat_name} **not** equal to {feat_value}, then **not** {label} | If szoj not equal to 3, then not 5 |
| quantifier | If {cond}, then it is **{quantifier}** {label} | If twqk equal to no, then it is seldom fem |

Table 4: Templates and example explanations for different task complexities in the synthetic training benchmark. For brevity, we omit mentions of combinations of complexities, e.g., conjunction + quantifier.

| Task | Explanations from CLUES-Real | MaNtLE-PF Explanations |
|---|---|---|
| banknote-authentication | Below 3.80 skewness leads to the original. | If skewness lesser than or equal to 3.049, then it is occasionally Fake. |
| | Kurtosis is high value so it is original. | If kurtosis lesser than or equal to 0.995, then it is often Fake
If kurtosis lesser than 9600, then it is frequently Fake × |
| indian-liver-patient | The SGPT Low percentage so the liver patient was no | If SGPT lesser than or equal to 39, then patient is generally No |
| | Age group above 40 ensures liver patient | If age lesser than or equal to 39, then patient is generally No |
| tic-tac-toe-endgame | Without b categories in middle middle square comes under the Positive group. | If middle-middle-square equal to x, then Game over is sometimes positive |

Table 5: Explanations generated by **MaNtLE** for CLUES-Real datasets.

datasets. While our training procedure learns to generate explanations for datasets akin to these prior works, our primary objective is to explain classifiers to understand their classification rationale rather than datasets.

## C OOD Synthetic Task Results

In §4.1, we evaluated **MaNtLE** explanations on a set of 20 unseen tasks from the synthetic benchmark. The results of this evaluation, presented in Figure 9, include **MaNtLE**'s performance on the full range of generation metrics, as well as measures of faithfulness and simulatability. As noted previously in §4.1, the performance metrics for generation, such as BERT-Score, ROUGE-*, and BLEU, have revealed that **MaNtLE** explanations exhibit a closer alignment to the ground-truth explanations when the latter includes quantifiers. This phenomenon can be attributed to a bias that **MaNtLE** acquires towards generating quantifiers towards the end of the training process, as observed in Table 1. However, our analysis of faithfulness and simulatability scores revealed that **MaNtLE** explanations were most effective on the simplest datasets that lacked complexities such as negations, quantifiers, or conjunctions, in line with our expectations.

### C.1 Comparison with WT5

In this section, we compare the effectiveness of a WT5-style model (Narang et al., 2020) over our proposed objective. In WT5, the explainer makes predictions for each example, and then we pick the best explanation that's most faithful to all the input examples as the one that explains the whole set.

| Method | Simulatability Score |
|---|---|
| WT5 | 55.50 |
| **MaNtLE** | 93.72 |

Table 6: Comparing the simulatability of explanations generated by WT5 and **MaNtLE** on a set of 20 unseen synthetic classification tasks.

To circumvent the lengthy pre-training process, which took approximately 25 days as detailed in Appendix §A.2, we opted to train a WT5-style model on the conjunctions-based dataset analyzed in Section §6.1. We fine-tuned both the WT5-style model and **MaNtLE** by training them on 48,000 tasks for 200K batches. After the fine-tuning stage, we evaluated the simulatability of explanations generated by each method on 20 held-out tasks. For **MaNtLE**, we directly evaluate the generated explanations on the subset of tasks. On the other hand, for WT5, we evaluated the most faithful explanation from the set of explanations generated for each example in a batch.

It is important to acknowledge that this represents just one approach to adapting the NLE baseline for explaining sets of examples. Nevertheless, we believe that the exploration of deriving overarching patterns from multiple single-example explanations poses intriguing research questions that warrant separate investigations.

From Table 6, we observe that **MaNtLE** achieves a higher simulatability score on the held-out tasks in comparison to the WT5 model, underscoring the generalization utility of our procedure to explain subsets of examples.

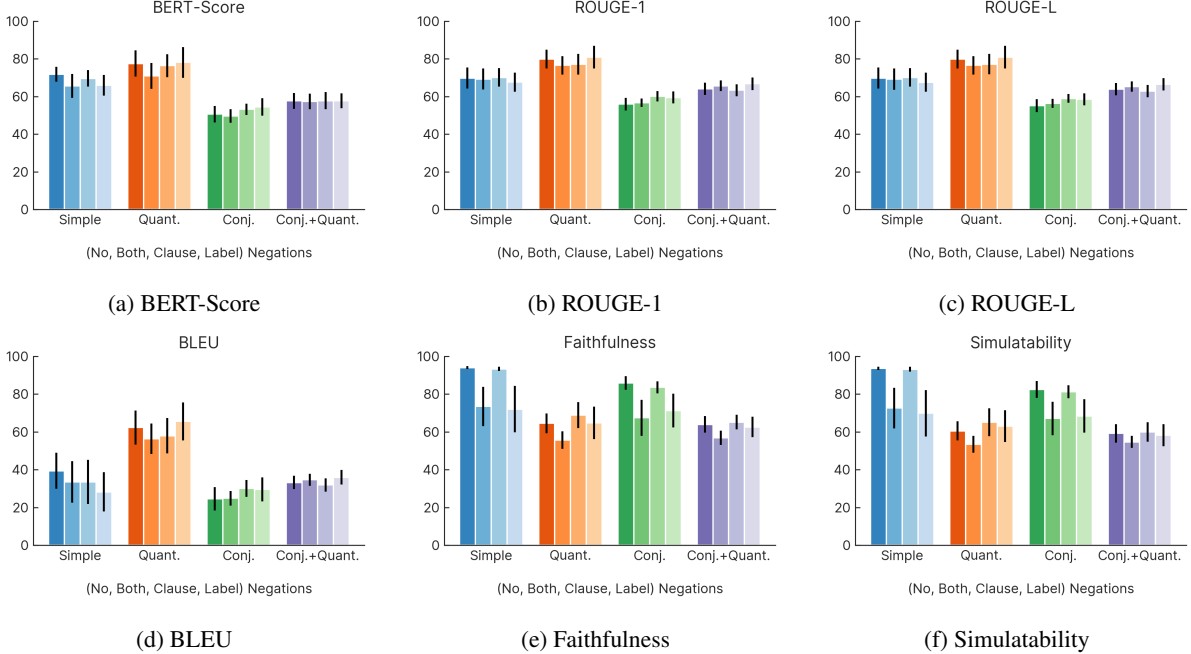

(a) BERT-Score        (b) ROUGE-1        (c) ROUGE-L

(d) BLEU        (e) Faithfulness        (f) Simulatability

Figure 9: Results on OOD tasks of different complexities (negations, quantifiers, conjunctions). These are numbers averaged over 20 datasets per task category. BERT-Score, ROUGE, and BLEU scores are the highest for Quantifier datasets since our model is more adept at generating content that contains single attribute explanations with quantifiers and negations.

## D  Extended Analysis

### D.1  Can MaNtLE take advantage of more examples?

Extending on the results from §6.2, which presented the performance of `logistic regression` and `decision tree` models trained on the `adult` dataset, we further investigate the performance of `neural network` and `xgboost` classifiers in this section. In addition, we evaluate the precision and coverage of the explanations on the simulation set (i.e., the test set). Consistent with our findings in §6.2, increasing the number of examples used by **MaNtLE** consistently improves the precision of the explanations, leading to improved overall performance (Figure 10).

### D.2  Can MaNtLE handle tasks with more features?

In this experiment, we evaluate the ability of **MaNtLE**, and the `PerFeat` decoding variant of **MaNtLE**, to generate explanations for logistic regression and decision tree classifiers as we increase the number of features from 5 to 11 for the `adult` dataset. As in the prior sections, we measure the faithfulness and simulatability metrics.

In §5, we experimented with exactly five features for all datasets. However, in real-world situations,

classifiers may operate over more than five features, which is why this evaluation is essential.

The results in Figure 11 suggest that the faithfulness of generated explanations is invariant to the number of input features used by a classifier that we seek to explain. However, while **MaNtLE**-PF generates more faithful explanations than **MaNtLE**, this advantage does not translate to improved simulatability as the number of features increases.

### D.3  Do predicted quantifiers reflect the precision of explanations?

We conducted an empirical study to investigate the relationship between quantifiers used in explanations generated by **MaNtLE** and the precision of the explanations upon execution. Specifically, we mapped the quantifiers used in the explanations, such as 'certainly', to their corresponding numeric values and analyzed the frequency of the predicted label values when the conditions mentioned in the explanations were met. For instance, if we have a generated explanation for the `adult` dataset of the form, *'if education equal to dropout, then income is certainly <=50K'*, where *'certainly'* maps to a value of 95%, we check if the label is '<=50K' 95% of the time when the education level was dropout on some held-out set of examples. As **MaNtLE** was

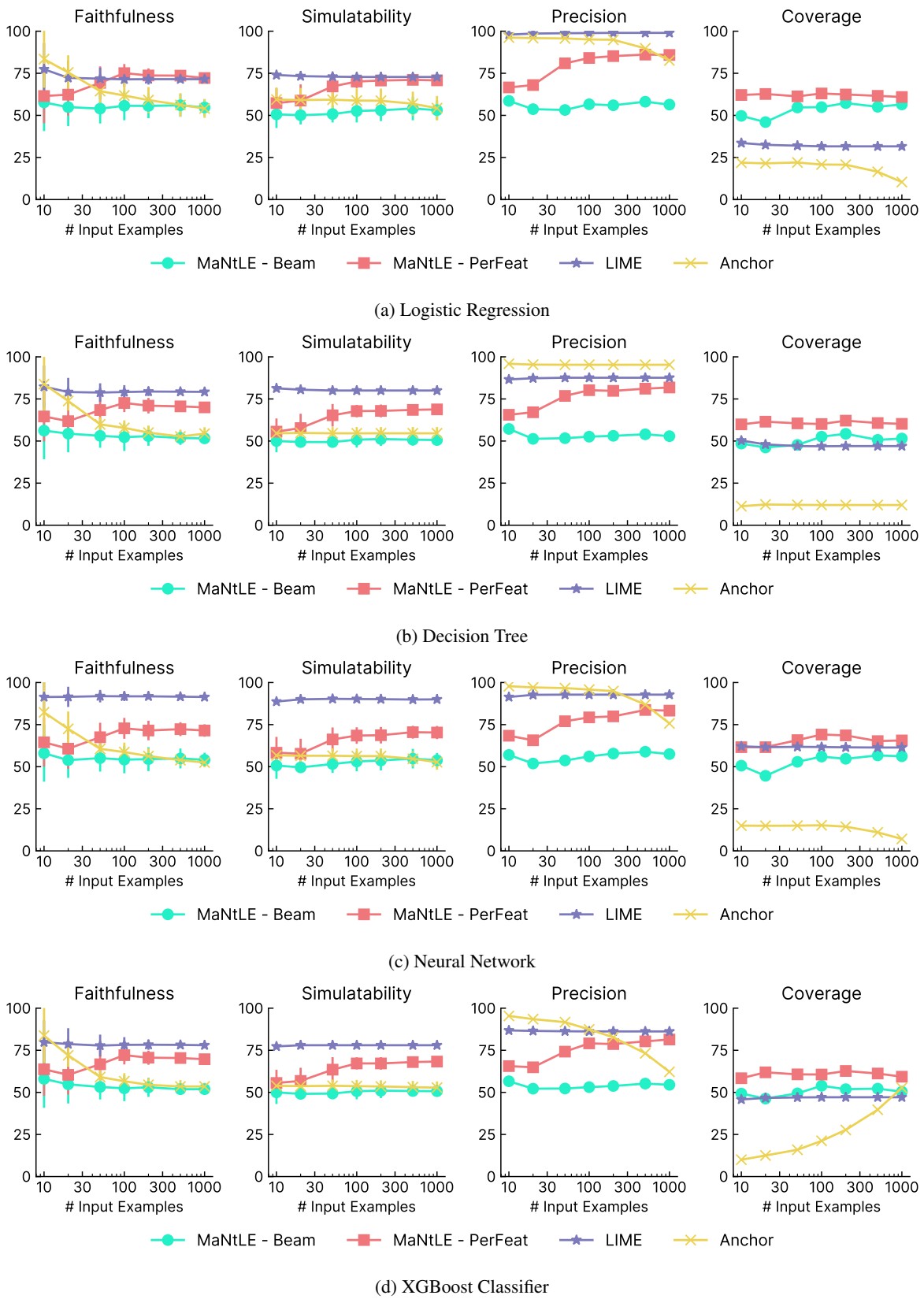

Figure 10: **Increasing number of input examples improves simulatability and faithfulness of MaNtLE-PF explanations.** Here, we show the results by increasing the number of input examples between 10 and 1000 for different classifiers trained on the Adult dataset. While the simultability of LIME, Anchor, and MaNtLE remain constant with an increase in the number of input examples, explanations from MaNtLE-PF improve simulatability and faithfulness.

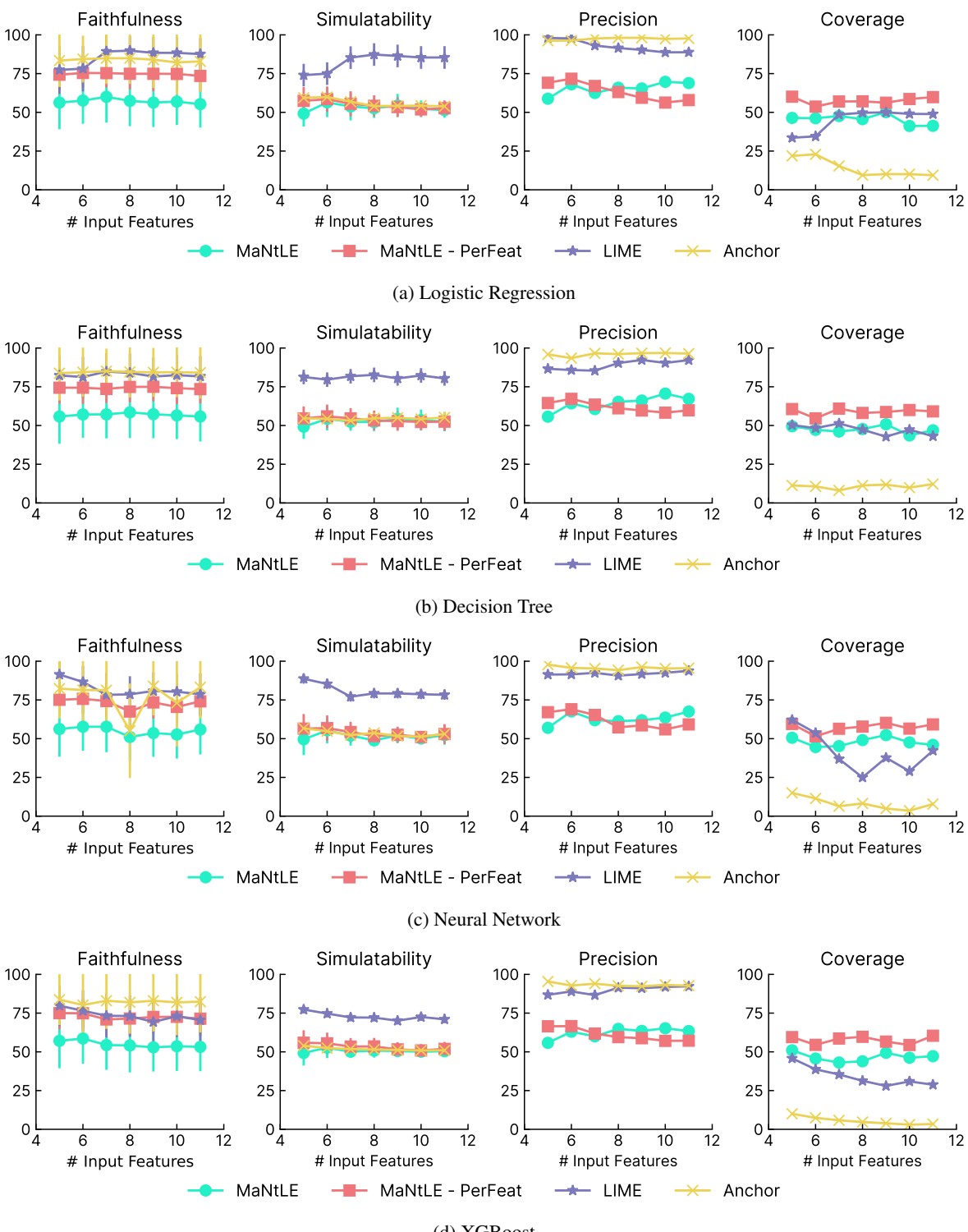

Figure 11: **Faithfulness of `MaNtLE`-PF explanations is largely independent of the number of input features across models**. Here, we compute metrics over 100 runs for the Adult dataset. The simulatability of explanations decreases with an increase in the number of features, as would be naturally expected.

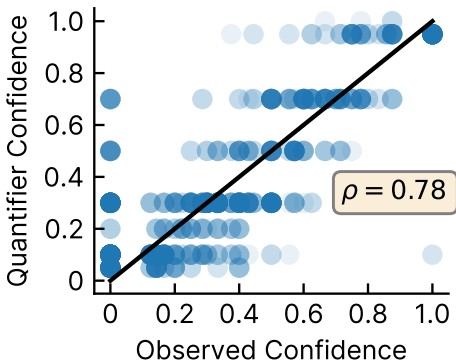

Figure 12: Quantifier predictions in `MaNtLE` explanations strongly correlate with the conditional frequency of outcomes in the underlying input.

trained on a synthetic dataset, whose quantifier datasets were constructed by mapping numeric values to words (using the map from (Srivastava et al., 2018)), we can convert the generated quantifiers back to numeric values and perform the analysis.

In Figure 12, we plot the correlation between the (predicted) quantifiers and the explanation precision for explanations generated from classifiers trained on the `adult` and `travel-insurance` datasets. We observe a strong correlation between the predicted quantifiers and precision of explanations (Spearman $\rho = 0.78$), indicating that the `MaNtLE` is able to generate accurate and reliable explanations of a black-box classifier's reasoning process.

# E  Human Evaluation

In this section, we elaborate on our human evaluation setup and provide templates as well as workers' accuracies in solving the adult task using different explanations.

**Evaluation details.** Firstly, during a pilot study, we identified that workers found it difficult to interpret the meanings of different quantifiers. As a result, following the confidence values in Srivastava et al. (2018), we reverse map the quantifiers in `MaNtLE` explanations to confidence values and present them to the turkers. For example, given *'if Education not equal to Dropout, then Income is certainly >50K'*, we convert it to *'95% of the time, the Income is >50K if Education not equal to Dropout'*. Secondly, workers preferred high-confidence explanations and often rated other explanations poorly. To make a fair evaluation of our explanations, we restrict human evaluations to settings where `MaNtLE` explanations are at least 85% confident of their ex-

planations (where confidence is measured by the quantifier used in the generated explanation). Finally, in experiments, we ask workers to simulate the behavior of the different classifiers used in our simulation experiments.

The templates used in our experiments can be found in Figure 13 (for perturbation and simulation experiments) and Figure 14 (for explanation preference experiments). Examples of explanations presented to workers can be found in Figure 15.

**Worker Accuracy.** Here, we present the classification accuracy (averaged over workers) during the pre-explanation and post-explanation phase of the perturbation and simulation experiments described in §5.4. It is worth noting that individual workers may have varying degrees of pre-explanation accuracy, thereby making a direct comparison of raw accuracies between explanation methods misleading. Our primary objective is to investigate whether explanations improve the workers' classification ability. Therefore, we depict the percentage of instances where the explanations led to improved classification performance in Figure 4.[10]

| Experiment | Exp. Method | Pre-Exp. Accuracy | Post-Exp. Accuracy |
|---|---|---|---|
| Perturbation | LIME | 71.5 | 67.7 (↓) |
| | Anchors | 63.1 | 55.4 (↓) |
| | **MaNtLE** | 53.8 | 60.8 (↑) |
| Simulation | LIME | 60.0 | 56.9 (↓) |
| | Anchors | 67.7 | 63.8 (↓) |
| | **MaNtLE** | 66.1 | 67.7 (↑) |

Table 7: Average classification accuracies of workers on perturbation and simulation experiments for the `adult` dataset in the pre-explanation and post-explanation phases. Overall, 23 workers took part in this study. Exp.= explanation

---

[10]When we mention the classification performance of human workers, we refer to the number of times they can predict the same label as the classifier, whose explanations they see during the learning phase.

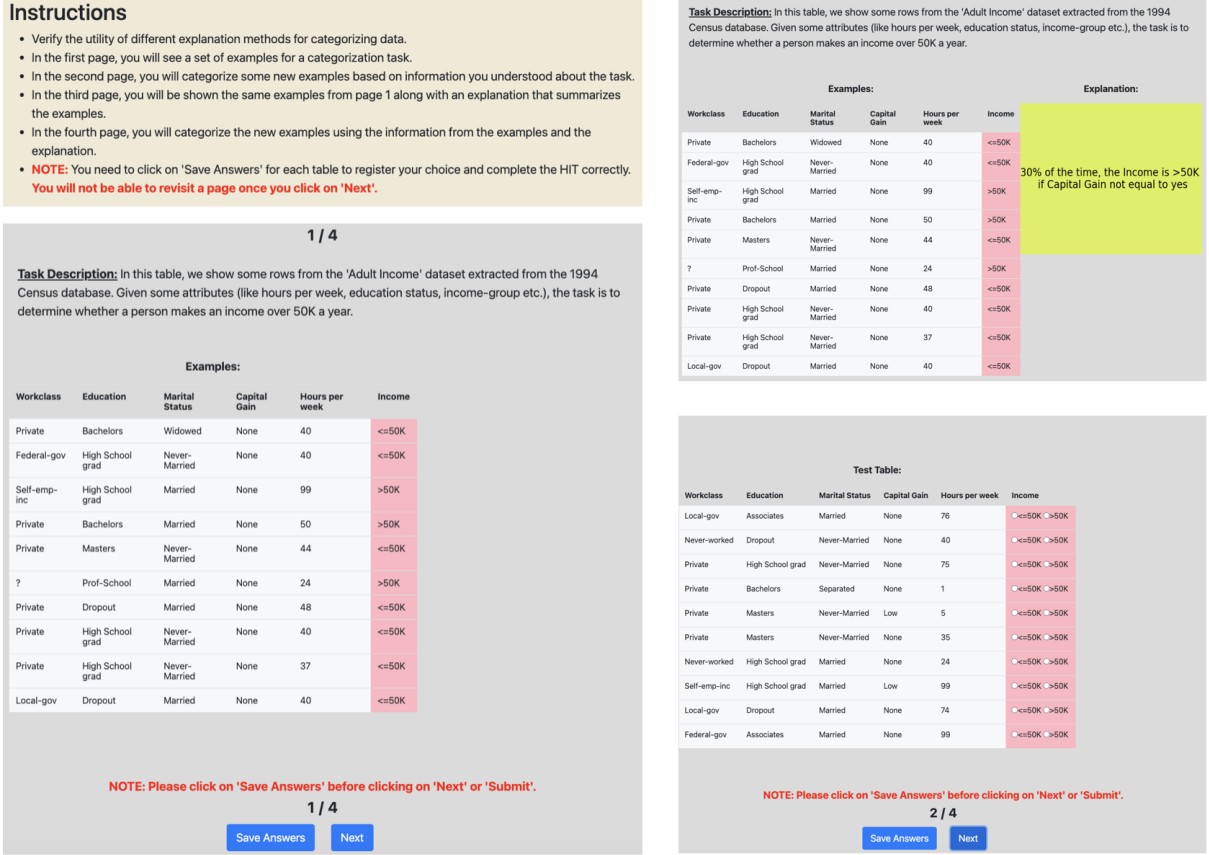

Figure 13: Template for performing the perturbation and simulation experiments using examples from the `adult` dataset.

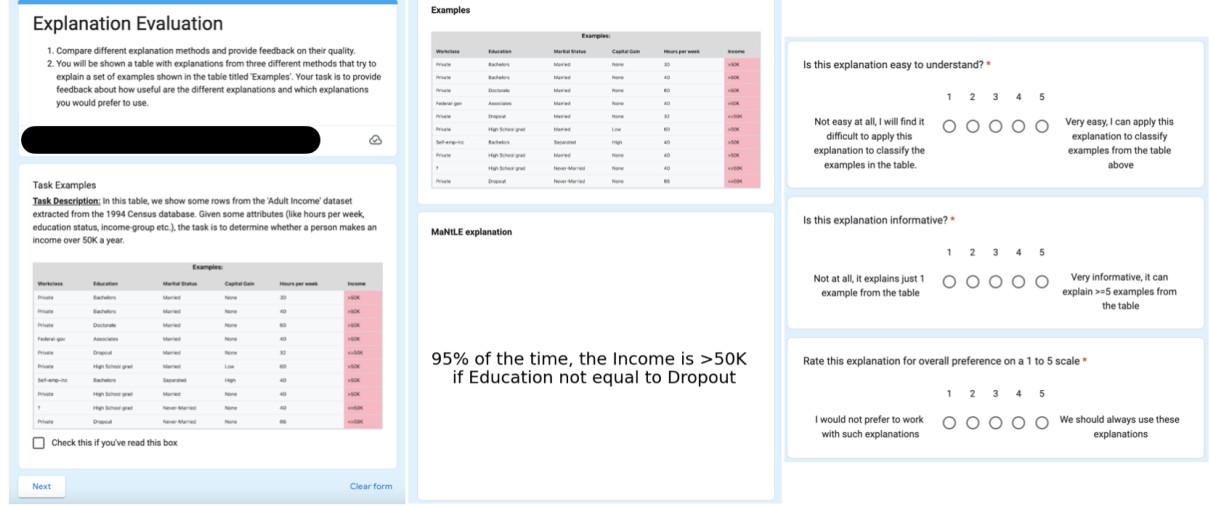

Figure 14: Template for performing the subjective evaluation of different explanations on a 1-5 Likert scale for understandability, informativeness, and overall preference.

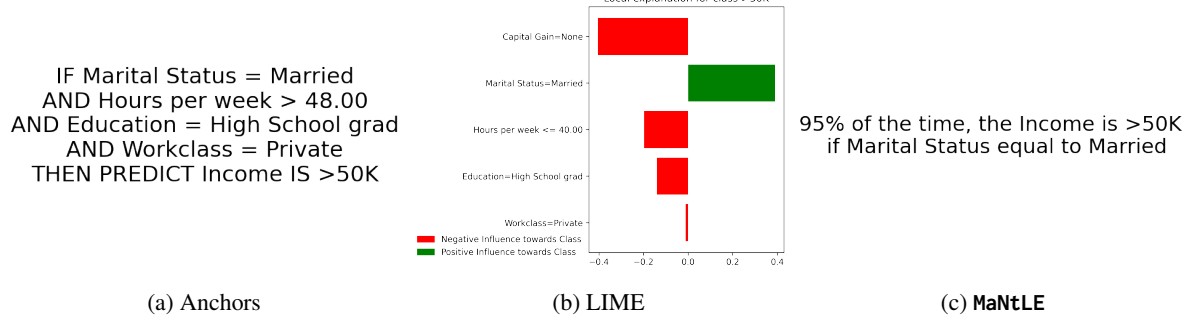

IF Marital Status = Married
AND Hours per week > 48.00
AND Education = High School grad
AND Workclass = Private
THEN PREDICT Income IS >50K

95% of the time, the Income is >50K
if Marital Status equal to Married

(a) Anchors             (b) LIME             (c) MaNtLE

Figure 15: Explanation format used in human evaluation experiments to investigate the faithfulness and simulatability of explanations generated by different methods.