# OpenReview forum: "MaNtLE: Model-agnostic Natural Language Explainer"
_EMNLP/2023/Conference — EMNLP 2023 Main_

### Official Review · Reviewer_CbXw · 2023-07-31

**Soundness:** 4

**Excitement:**

4: Strong: This paper deepens the understanding of some phenomenon or lowers the barriers to an existing research direction.

**Paper Topic And Main Contributions:**

In this work, the authors propose to use large language models to generate explanations for classifier predictions.
The proposed method is simple yet effective.
Specifically, this work fine-tunes a T5 model on the thousands of synthetic classification tasks (with natural language explanations). Then, the learned model is used to generate explanations for unseen tasks.
The synthetic tasks and explanations are completely automatically generated, which saves a lot of human effort comparing previous methods.

This work first compares the generated explanations to the human-written explanations on the CLUES-Real benchmark. The results show that the proposed method can generate explanations that are similar quality of human explanations.
Then,  the proposed method is compared against previous works, namely LIME and Anchors. The results show that the proposed method is better than the baselines in terms of faithfulness and simulatability

**Questions For The Authors:**

- Do you have the online demo? I am interested to see the demos.
- Discussion about scaling up to hundreds of attributes? I have the impression that this is mainly limited by the maximum sequence length of transform models.
- Can this method be extended to embedding features?
- Does the synthetic datasets include noise? I think this might be important to strengthen the robustness of the model.

**Reasons To Accept:**

- Simple and effective method.  This method provides a new direction for the language explanations for model predictions, which does not need any human labeling effort. With the rapid development of LLM, I think there are a lot of research ideas to explore in this direction.
- Well-written paper. Easy to follow.

**Reasons To Reject:**

- Based on the categorization of explainability methods discussed in Section 2, I am expecting to compare the proposed method against language explanation methods, such as CAGE or WT5, instead of feature attribution methods, such as LIME and Anchor.  I have concerns about these mismatched baselines.


**Reproducibility:**

3: Could reproduce the results with some difficulty. The settings of parameters are underspecified or subjectively determined; the training/evaluation data are not widely available.

**Reviewer Confidence:**

3: Pretty sure, but there's a chance I missed something. Although I have a good feel for this area in general, I did not carefully check the paper's details, e.g., the math, experimental design, or novelty.

---

> ### Author Rebuttal · Authors · 2023-08-28
>
> We thank the reviewer for their motivating comments on our draft. We are particularly grateful for the acknowledgment of the “simplicity” and “effectiveness” of our work alongside the appreciation for the readability of our work. Accessibility of our work to readers from diverse expertise is paramount to us; hence, this acknowledgment is particularly encouraging. Below, we address some of the queries from the reviewer:
> - **NLE baselines**: Kindly refer to the point mentioned under Reviewer P3Ea (1st review on this page) with the same sub-title.
>
>
> - **Online demo**: Thanks for the suggestion! We will create a website for hosting the demo to improve accessibility and interaction for other researchers.
>
>
> - **Scaling to thousands of attributes**: Indeed, as the reviewer mentions, we are limited by the number of tokens that can be used as input to the transformer model. Incorporating techniques like ALiBi (Press et al. 2022) to extend the inference time input for models would be an exciting future direction to stretch the application of MaNtLE for this scenario.
>
>
> - **Extending to embedding-based features**: If the point of discussion is shifting the attribute values alone from text to embeddings, it may yet be possible to train such a model for categorical tasks. However, suppose we are to replace the structured format altogether with embeddings. In that case, the model may be untrustworthy as we may not completely understand what sections of the input contributed to the explanation.
>
>
> - **Do synthetic datasets include noise?** It would be nice to gain some more clarity on the kind of noise referenced here. Through the use of quantifiers in the synthetic task explanations, we are already introducing some stochasticity in the predictions. Further, we randomize the order of features in every training batch to ensure that the order of feature names does not bias the generated explanations.
>
>
> **References**:
> - Press et al. 2022 : Train Short, Test Long: Attention With Linear Biases Enables Input Length Extrapolation, ICLR 2022

---

### Official Review · Reviewer_tLzw · 2023-08-05

**Soundness:** 3

**Excitement:**

3: Ambivalent: It has merits (e.g., it reports state-of-the-art results, the idea is nice), but there are key weaknesses (e.g., it describes incremental work), and it can significantly benefit from another round of revision. However, I won't object to accepting it if my co-reviewers champion it.

**Missing References:**

A lot of papers on NLE are not cited.

**Paper Topic And Main Contributions:**

This paper proposes a new explainability tool (MaNtLE) for machine learning systems. Their explainer targets on explaining structure classification tasks with natural language explanations (NLE). To build such a classifier, they fine-tune T5-large on thousands of synthetic classification tasks with NLE supervision using multi-task training. They evaluated the usefulness of explanations by measuring the model classification-utility of the generated explanations as well as human evaluation (including understandability, informativeness, and overall preference). They showed the MaNtLE outperforms previous interpretability tools LIME and Anchors.

**Questions For The Authors:**

A. Why are the classification accuracy so low in Table 2? The highest number is 63% on the binary classification dataset. Additionally, where does the 7% on L333 come from?

B. Why do you sample 100 subsets (each with 10 examples) for automatic evaluation? Why do you choose it over having less subsets but each with more examples?

C. Why are there only 10% examples that belong to `{label}` and `not {label}` classes? What are the other classes? I thought all non-target-label classes are already converted to `not {label}` class.

**Reasons To Accept:**

- Proposes a model-agnostic explainer that generates natural language explanations for structured classification tasks.
- Evaluated their method extensively: including comparing to popular interpretability tools, conducting both automatic evaluation and human evaluation, and further analyzing the factors affecting the generalizability of MaNtLE.

**Reasons To Reject:**

- Unfair comparisons to baselines: The authors trained T5-large on thousands of synthetic tasks, while only limiting LIME and Anchors to use very few perturbations / training examples. They acknowledged that “LIME uses 5k perturbations to fit a good linear model and generated explanations for tabular datasets.” So it is unclear to me whether their results in showing MaNtLE outperforming LIME is because their method is actually effective or because they simply did not provide enough input to LIME.
- Missing NLE baselines: The authors only compared to feature attribution type of explanations, but their explainer is generating natural language explanations. They should compare with other NLE interpretability tools.
- Evaluation is not different from NLEs: The authors defined faithfulness and simulatability as classification utility on training (perturbation) and testing set, respectively. This is different from what they claim to be doing differently from previous works that focuses on improving model classification performance with natural language explanations. The authors should seek to evaluate how well MaNtLE can explain model behavior (as claimed in related work to differentiate themselves from the other NLE work), but the current two metrics do not show how they are measuring this.
- Unclear automatic evaluation for LIME and Anchors: Unclear how LIME and Anchors explanations are being used to predict the label. In particular, LIME generates attribution scores for each feature, so it is not a natural language explanation format. If the authors use T5 to predict the label, then it is unfair because T5 is a LM, which handle natural language input better.

**Reproducibility:**

3: Could reproduce the results with some difficulty. The settings of parameters are underspecified or subjectively determined; the training/evaluation data are not widely available.

**Reviewer Confidence:**

3: Pretty sure, but there's a chance I missed something. Although I have a good feel for this area in general, I did not carefully check the paper's details, e.g., the math, experimental design, or novelty.

**Typos Grammar Style And Presentation Improvements:**

Typo:

- Missed underscore for M in title.
- Conventionally, figures should be on the top of the page. Figure 2 is currently below a footnote.

---

> ### Author Rebuttal · Authors · 2023-08-28
>
> We thank the reviewer for their valuable comments on our draft. We are grateful for the reviewer’s acknowledgment of the “extensive evaluation” of MaNtLE in our work. Below, we address these concerns raised by the reviewer:
>
> - **Unfair comparison with LIME/Anchors**: Our evaluations in Section 5 are geared towards fair computations for each method on the new datasets that have been presented. From the perspective of computation utilized for the new task, the baselines aren’t the only budget-constrained approaches; MaNtLE is budget-constrained, too. To the best of our knowledge, we have applied the LIME and Anchors to their best ability (barring the model call computation) and hence find it a completely fair evaluation of approaches.\
> \
> It is important to note that in Section 6.2, the experiment displayed LIME and Anchors with full budget allocations, granting LIME access to 5000 perturbations and Anchors access to the entire training set. Consequently, Figure 7 portrays higher numbers of faithfulness and simulatability for LIME. However, MaNtLE remains restricted to the budget constraint, and therefore has lower numbers. Nevertheless, as the number of examples given to each method as input increases, the performance of MaNtLE-PF improves, and closely matches that of LIME.
>
>
> - **NLE Baselines**: Please refer to the point under Reviewer P3Ea (1st review on this page) with the same sub-title.
>
>
> - **Evaluation not different from NLEs**: We feel there is a misunderstanding here. MaNtLE is utilized to explain classifiers per our original goal indicated throughout the paper. Concretely, the experiments in Section 5, 6.2, and Appendix D seek to explain different classifiers (logistic regression, neural networks, decision trees, and XGBoost) trained on three different datasets.\
> \
> In faithfulness and simulatabilty experiments, we evaluate the effectiveness of MaNtLE’s explanations in retrieving the same predictions as made by the classifiers for examples from the input (for faithfulness) and unseen examples (for simulatability). The unseen examples could have been from the (unseen) training, validation, or test set. Our choice of using the test set for simulatability in that sense is arbitrary and bears no relevance to the final conclusions.
>
>
> - **Unclear automatic evaluation of LIME/Anchors**: Firstly, we would like to re-iterate that MaNtLE does not perform any classification (see Figure 1). It explains the predictions made by other classifiers. Aside from T5 forming the base of MaNtLE, we do not explain a T5 model’s predictions anywhere in the paper.\
> \
> Now, we would like to clarify the reviewer’s query wrt the evaluation of LIME. LIME (Ribeiro et al. 2016) fits a linear regression module (over feature values) that predicts the probability of a particular example belonging to {label} vs. not {label}. The weights of the linear classifier are used as the feature attributions, as per the original paper. Now, to make predictions using the LIME explanations, we merely use these feature attributions as weights of a linear classifier again and predict on new examples. A similar logic flows for Anchors as well. We hope this clarifies the doubt here.
>
>
> - **Low classification accuracies in Table 2**: Human performance using crowdsourced explanations for these datasets is estimated to be 70% (as per Menon et al. 2022). Hence, assuming some error due to incomplete natural language understanding, having classification accuracies in the ballpark of 60% is expected for models like LaSQuE (Ghosh et al. 2023). \
> \
> Regarding the 7% gap mentioned in L333, we regret the confusion caused by this error. We meant that the gap between the *best* MaNtLE explanations and the crowdsourced explanations is within 4% of each other (absolute).
>
>
> - **100 subsets**: The concept of utilizing 100 subsets is analogous to conducting 100 distinct model runs with varied seeds, except that in this case, we evaluate 100 diverse subsets of input data. This was done to provide a more robust estimate of the different metrics of interest in our paper.\
> \
> It's important to mention that the experiment requested by the reviewer has been executed and is detailed in our analysis within Section 6.2. It's noteworthy that the inclusion of additional strategies is essential due to the inherent limitation that not all examples can fit into MaNtLE's input due to restrictions on the maximum token length. For details, please refer to the elaboration in Section 6.2.
>
> - **On 10% examples belonging to label and non-label classes**: We would like to clarify that the statement mentions “at least 10% of the examples belong to either class” (L241-242). Accordingly, all examples passed into MaNtLE are either of the ‘{label}’ or ‘not {label}’ class with *AT LEAST* 10% of each.
>
> **References**:
> - Ribeiro et al. 2016: “Why Should I Trust You?” Explaining the Predictions of Any Classifier,  SIGKDD 2016
> - R. Menon et al. 2022 : CLUES: A Benchmark for Learning Classifiers using Natural Language Explanations, ACL 2022
> - Ghosh et al. 2023: LaSQuE: Improved Zero-Shot Classification from Explanations Through Quantifier Modeling and Curriculum Learning, Findings of ACL 2023

---

### Official Review · Reviewer_vjwc · 2023-08-06

**Soundness:** 3

**Excitement:**

3: Ambivalent: It has merits (e.g., it reports state-of-the-art results, the idea is nice), but there are key weaknesses (e.g., it describes incremental work), and it can significantly benefit from another round of revision. However, I won't object to accepting it if my co-reviewers champion it.

**Missing References:**

Perhaps relevant, work finding semantically-coherent subsets models do poorly on (e.g., https://arxiv.org/abs/2203.14960 and followups).

**Paper Topic And Main Contributions:**

This paper introduces a new technique to explain model behavior, specifically focusing on explaining a subset of points (as opposed to individual point). The method works by training a T5 model on thousands of synthetic tasks for which an explanation of the labels is known; specifically, they fine-tune a model to predict the explanation (via generation) from 10+ examples with labels. In order to come up with a new explanation, the authors feed in examples (with the constraint that at least 10% of the examples come from each task), then either greedily generates an explanation, generates 20 with a beam search (then chooses the one that most faithfully follows the labels on the chosen points), or generates 20 using a method that favors diversity, then chooses the most faithful again. The authors find that their method generalizes reasonably well to unseen synthetic datasets, outperforms Anchor and LIME on adult, but largely loses out to Anchor on travel insurance and recidivism (even in the “budget-constrained” setting where Anchor gets much fewer samples than general), and in a human study finds that their (postprocessed) explanations are more useful for humans to predict model behavior than those from LIME or anchors (on the adult dataset).

**Questions For The Authors:**

* Could you provide random examples of explanations generated by each method on Adult, Travel Ins., and Recidivism?
* On more realistic tasks, how can you run MaNtLE-BS and MaNtLE-PF (to evaluate faithfulness)
* Is there a difference between explaining a subset of the data versus the entire predictor?

**Reasons To Accept:**

* The paper’s approach of fine-tuning explanations over many different tasks seems interesting, and very novel relative to prior work.
* The authors study explanations in a range of settings (synthetic, zero shot generation, human in the loop, etc.) which I appreciated.
* Their method, MaNtLE, is very efficient at inference time; unlike existing instance-level methods that require perturbing an input in order to come up with an explanation, MaNtLE simply generates explanations

**Reasons To Reject:**

* MaNtLE to do worse on many of the realistic tasks presented, despite some advantages (get to choose the best of 20 explanations, other methods are budget constrained, while mantle sees thousands of synthetic tasks, extra postprocessing for mantle explanations instead of LIME).
* The authors frame MaNtLE as acting on subsets, but it seems like it’s aiming to evaluate full (via simulatability metric). This means MaNtLE likely misses all tail behavior / doesn’t produce “correct” explanations. And it’s unclear if the MaNtLE explanations are more faithful than just the standard label function (even when the model differs from this due to imperfect accuracy
* MaNtLE-BS and MaNtLE-PF rely on having a way to quickly test the faithfulness of explanations, which seems intractable on real datasets.
* Correctness of explanations is measured “functionally”, but if the human (or zero-shot model) isn’t getting perfect accuracy, the explanations are still wrong.
* It would be helpful to have more motivating examples for what counts as an explanation (and how to parse it) for the synthetic data.

**Reproducibility:**

3: Could reproduce the results with some difficulty. The settings of parameters are underspecified or subjectively determined; the training/evaluation data are not widely available.

**Reviewer Confidence:**

4: Quite sure. I tried to check the important points carefully. It's unlikely, though conceivable, that I missed something that should affect my ratings.

**Typos Grammar Style And Presentation Improvements:**

Typos
* Line 211: comma error

---

> ### Author Rebuttal · Authors · 2023-08-28
>
> We thank the reviewer for their valuable comments on our draft. We are particularly encouraged that the reviewer finds our work “very novel” and our method “very efficient at inference”. Additionally, we are thankful for the reviewer’s “appreciation” of the “wide range of [experimental] settings” in our work. The reviewer also raised some thoughtful concerns about our work. Below, we address these concerns:
>
> - **MaNtLE sees thousands of synthetic tasks; Baselines are budget-constrained**: We would like to clarify that pre-training is done just once, and the model is never fine-tuned on new tasks. It's important to note that this pre-trained model will be released upon publication. Thus, pre-training is not an additional demand placed on the user of MaNtLE.
> Our evaluations in Section 5 are geared towards fair computations for each method on the new datasets that have been presented. From the perspective of computation utilized for the new task, the baselines aren’t the only budget-constrained approaches; MaNtLE is budget-constrained, too. To the best of our knowledge, we have applied the LIME and Anchors to their best ability (barring the model call computation) and hence find it a completely fair evaluation of approaches. We can provide a more detailed explanation if the reviewer could kindly articulate their exact concern with respect to computation.
>
>
> - **MaNtLE acts on subsets but evaluates on full**: Note that MaNtLE is not optimized to maximize simulatability. This is an emergent attribute from large-scale pre-training. Ideally, we would like to obtain explanations from MaNtLE that can be applied across all model predictions. However, it is possible that MaNtLE misses out on explaining some tail behavior, as pointed out by the reviewer. Conceptually, this can be alleviated by recursively applying MaNtLE to explain different subsections of the data to improve the coverage of explanations.
>
>
> - **Examples of explanations in Adult/Recidivism/Travel-Insurance**: We would like to point out that Figure 15 (in the Appendix) provides an example of the explanations from the three methods. However, given the graphical nature of LIME explanations (Figure 15), we instead present some of MaNtLE-PF’s explanations explaining the logistic regression classifier in results of Section 5 below:\
> \
> Adult:
> 1. If Hours per week equal to 40, then Income is often >50K
> 2. If Marital Status equal to Married, then Income is definitely >50K
> 3. If Education equal to High School grad, then Income is sometimes >50K\
> \
> Recidivism:
> 1. If Alcohol not equal to Yes, then Recidivism is occasionally commit
> 2. If Race equal to White AND Priors equal to 1 to 5, then Recidivism is likely commit
> 3. If PrisonViolations not equal to Yes, then Recidivism is seldom commit\
> \
> Travel-insurance:
> 1. If FrequentFlyer not equal to Yes, then the customer is never interested in travel insurance
> 2. If AnnualIncome not equal to Low, then the customer is certainly interested in travel insurance
> 3. If Age greater than 26, then the customer is typically interested in travel insurance\
> \
> More examples of explanations generated by all methods will be incorporated in the future version.
>
> - **Intractability of faithfulness/simulatability computation on real-world datasets**: We assume that the question is in relation to the use of a semantic parser in our experiments here. In more general scenarios, where we can have more diverse linguistic elements in explanations, we can assume access to models like LaSQuE (Ghosh et al. 2023) to provide approximate estimates of the utility of different explanations. We also note that we can parallelize the computation of these metrics for each explanation, making this time-efficient.
>
> - **Difference between explaining subset vs. entire predictor**: Great question! Suppose we are interested in understanding how the educational qualifications of individuals in a specific province (let's call it "Province X") in a country ("Y") influence their earnings. However, we are handed a national database containing information from across the country.  In this situation, our approach would be to filter out the “subset” of examples corresponding to people in province X from the national database and run MaNtLE to interpret the general pattern of educational background among people in the province. Of course, this is a toy scenario trying to explain information from a dataset (or database). A similar analogy can be made to analyze model predictions on different sections of data.
>
> **References**:
> - Ghosh et al. 2023: LaSQuE: Improved Zero-Shot Classification from Explanations Through Quantifier Modeling and Curriculum Learning, Findings of ACL 2023

---

### Official Review · Reviewer_P3Ea · 2023-08-07

**Soundness:** 3

**Excitement:**

2: Mediocre: This paper makes marginal contributions (vs non-contemporaneous work), so I would rather not see it in the conference.

**Paper Topic And Main Contributions:**

MaNtLE is a model-agnostic natural language explainer. It is fine-tuned T5-Large model on thousands of synthetic classification tasks, each paired with natural language explanations.

**Questions For The Authors:**

1. no other natural language explanation method is compared with?
2. the datasets compated (UCI income, recidivism, travel-insurance) are all tabular datasets? I dont see a strong case that in those datasets, natural language explanations would be better than attributon based method. If so, what is the main reason for that? also, for tabular dataset, if I have an attribution based method, it would be very easy to convert it into natural language based explanations.


**Reasons To Accept:**

1. the way constructing a synthetic dataset is interesting and smart. Though the setting is still artificial and simple.
2. comprehensive experiments.
3. the evaluation on CLUES-REAL and LaSQuE is interesting.

**Reasons To Reject:**

1. the experiments seem simple.
2. it would be better to include more discuss on in which cases natural language explanations would be more helpful. I dont think tabular dataset would make a strong case of this.

**Reproducibility:**

2: Would be hard pressed to reproduce the results. The contribution depends on data that are simply not available outside the author's institution or consortium; not enough details are provided.

**Reviewer Confidence:**

3: Pretty sure, but there's a chance I missed something. Although I have a good feel for this area in general, I did not carefully check the paper's details, e.g., the math, experimental design, or novelty.

**Typos Grammar Style And Presentation Improvements:**

I would prefer fidelity over faithfulness since the later is overloaded and controversial.

---

> ### Author Rebuttal · Authors · 2023-08-28
>
> We thank the reviewer for their valuable comments on our draft. We are particularly encouraged that the reviewer finds our experimentation and evaluation “comprehensive” and “interesting”. Below, we address some of the concerns raised by the reviewer:
>
> - **Natural Language Explanation (NLE) Baselines**: This is a question raised by other reviewers as well (reviewers:  P3Ea, tLzw, CbXw). We appreciate an opportunity to address this and clarify two fundamental points (which make these improper comparisons, and which guided our decision not to include these baselines):
>    - **MaNtLE explains other classifiers, rather than performing classification unlike other NLE baselines**: With NLE baselines, such as CAGE or WT5, the classifier jointly predicts ***both*** a label and a supporting explanation. The primary contribution of these methods is the induction of knowledge in explanations to improve classification. \
> \
>    In contrast, MaNtLE, as well as methods like LIME or Anchors, operates separately from the label-predicting classifier. It functions as a post-hoc explanation technique, deliberately dissociating the processes of classification and explanation. Hence, MaNtLE can be applied in a model-agnostic fashion for any classifier. The primary function of these methods is to explain classifiers rather than improve them. This distinction also accounts for the absence of LIME or Anchors as baselines in prior NLE works. (Rajani et al. 2019, Narang et al. 2020)
>
>    - **Explaining Subset of Examples vs. Single-example Explanations**: The focus of CAGE and WT5 is to explain the predictions made by a classifier for a *single-example*. In contrast, MaNtLE explains *subsets* of examples and extracts more general patterns. To compare fairly with MaNtLE, we require a method that can aggregate multiple single-example predictions. Deducing general patterns from multiple single-example explanations raises interesting research questions that can be a separate avenue of exploration. However, addressing this aspect goes well beyond the scope of our current study, and would overload the introduction of new methods.
>
>
> - **When and why would MaNtLE perform better and be preferred over attribution-based methods on tabular datasets?** As mentioned in 416-429, LIME and Anchors require access to predictions from the model for multiple perturbations of an example. However, if the classifier we applied for a particular tabular dataset were GPT-3, then querying this classifier would have a substantial monetary and environmental cost. In settings that necessitate the control of model calls, results in Table 3 conclusively indicate that MaNtLE would be the preferred method for explaining classifiers. \
> \
> Further, suppose we have tasks with some inherent noise in concept occurrence. In that case, MaNtLE can accurately capture such ambiguity by using quantifiers unavailable in attribution-based methods. \
> \
> Finally, we would like to point out that natural language explanations for tabular datasets are essential for the accessibility of interpretability tools to people from diverse domains (e.g., medical doctors) who may find it challenging to interpret explanations provided in the form of attribution scores.  Similar requirements for natural language explanations have been noted in Lakkaraju et al. (2022).
>
>
> - **Fidelity over faithfulness**: Thanks for the suggestion! We will take this into account and update the next version appropriately.
>
> **References**:
> - Rajani et al. 2019: Explain Yourself! Leveraging Language Models for Commonsense Reasoning, ACL 2019
> - Narang et al. 2020: WT5?! Training Text-to-Text Models to Explain their Predictions, arxiv 2020
> - Lakkaraju et al. 2022 : Rethinking Explainability as a Dialogue: A Practitioner’s Perspective, arxiv 2022

---

### Meta-Review · Area_Chair_UBnG · 2023-09-17

**Recommendation:** 3

**Metareview:**

This paper proposes MaNtLE, a model-agnostic natural language explainer which generates explanations for subsets of examples in structured classification tasks. According to the initial reviews, this paper conducts comprehensive experiments (studying explanations in several settings and performing both automatic and human evaluations). Also, the method is efficient at inference time without requiring input perturbation. However, there were some concerns raised, especially on the evaluation part (e.g., the missing NLE baselines and the unfair comparison to the baselines). During the rebuttal phase, the authors well addressed several questions and concerns of the reviewers. In particular, the authors argued that thousands of synthetic tasks MaNtLE saw (which seem unfair to other baselines) were used for pre-training MaNtLE, which happened only once, and this pre-trained model will be released upon publication so public users can leverage it in an efficient way. The authors also presented additional results comparing MaNtLE to a WT5-style model, served as an NLE baseline. Both models were pre-trained on the same number of synthetic tasks, and MaNtLE still outperformed WT5 in terms of simulatability. It has to be noted that, according to the authors' argument, MaNtLE and WT5 are not directly comparable by nature as the former explains a set of examples while the latter explains an individual example. The newly presented results were possible due to a simple aggregation method the authors used to enable the comparison.

---

### Decision · Program_Chairs · 2023-10-07

**Decision:**

Accept-Main

**Comment:**

This paper proposes MaNtLE, a model-agnostic natural language explainer which generates explanations for subsets of examples in structured classification tasks. According to the initial reviews, this paper conducts comprehensive experiments (studying explanations in several settings and performing both automatic and human evaluations). Also, the method is efficient at inference time without requiring input perturbation. However, there were some concerns raised, especially on the evaluation part (e.g., the missing NLE baselines and the unfair comparison to the baselines). During the rebuttal phase, the authors well addressed several questions and concerns of the reviewers. In particular, the authors argued that thousands of synthetic tasks MaNtLE saw (which seem unfair to other baselines) were used for pre-training MaNtLE, which happened only once, and this pre-trained model will be released upon publication so public users can leverage it in an efficient way. The authors also presented additional results comparing MaNtLE to a WT5-style model, served as an NLE baseline. Both models were pre-trained on the same number of synthetic tasks, and MaNtLE still outperformed WT5 in terms of simulatability. It has to be noted that, according to the authors' argument, MaNtLE and WT5 are not directly comparable by nature as the former explains a set of examples while the latter explains an individual example. The newly presented results were possible due to a simple aggregation method the authors used to enable the comparison.